# Position: Spectral GNNs Are Neither Spectral Nor Superior for Node Classification

## Abstract

Spectral Graph Neural Networks (spectral GNNs) for node classification promise frequency-domain filtering on graphs, yet rest on flawed foundations. Recent work shows that graph Laplacian eigenvectors do not in general have the key properties of a true Fourier basis, but leaves the empirical success of spectral GNNs unexplained. We identify two theoretical glitches: (1) commonly used "graph Fourier bases" are not classical Fourier bases for graph signals; (2) $(n-1)$-degree polynomial polynomials ($n$ = number of nodes) can exactly interpolate any spectral response via a Vandermonde system, so the usual "polynomial approximation" narrative is not explanatory. The effectiveness of GCN is commonly attributed to spectral low-pass filtering, yet we prove that low- and high-pass behaviors arise solely from message-passing dynamics rather than Graph Fourier Transform–based spectral formulations. We then analyze two representative directed spectral models, MagNet and HoloNet. Their reported effectiveness is not spectral: it arises from implementation issues that reduce them to powerful MPNNs. When implemented consistently with the claimed spectral algorithms, performance becomes weak. This position paper argues that: **for node classification, spectral GNNs neither meaningfully capture the graph spectrum nor reliably improve performance;** competitive results are better explained by MPNN equivalences, sometimes aided by incorrect implementations.

## 1. Introduction

Graph Neural Networks (GNNs) for node classification encompass several active research branches, such as Message Passing Neural Networks (MPNNs) (Gilmer et al., 2017; Rossi et al., 2024) and Spectral Graph Neural Networks (spectral GNNs) (Zhang et al., 2021a; He et al., 2021). Spectral GNNs form a niche branch rooted in spectral graph theory (Chung, 1997) and polynomial approximation techniques (Hammond et al., 2009). However, their polynomial filters are problematic in that performance does not reliably improve as filter order $K$ increases (He et al., 2021). A recent position paper (Guo et al., 2025) further argues that graph Laplacian eigenvectors should not be treated as a canonical Fourier basis in the classical sense, thereby challenging the usual "spectral" interpretation. We agree with this conclusion. However, this limitation should not be attributed to a continuous-versus-discrete mismatch claimed by Guo et al. (2025): Fourier representations extend naturally to discrete settings via the DFT (Oppenheim et al., 1999; Gonzalez & Woods, 2017). Instead, the gap stems from structural differences between Euclidean domains and general graphs—most notably the absence of a global shift/translation operator—so the Laplacian eigenbasis lacks some of the canonical properties that underpin Fourier analysis in regular domains.

**We take a stronger stance: spectral GNNs are theoretically flawed with several pitfalls, and their seemingly strong empirical performance stems primarily from equivalences to simpler message-passing neural networks (MPNNs), rather than from any genuinely spectral mechanism.**

### 1.1. Structure of this Position Paper

**Section 2:** We prove that Graph Fourier Basis $\neq$ Fourier Basis of Graph, exposing our first theoretical concern.

**Section 3:** Even granting that graph Fourier bases qualify as true Fourier bases, polynomial approximations fail on their own terms. Following spectral GNN reasoning, we show that on an $n$-node graph, an $(n-1)$th-order polynomial exactly interpolates any ideal spectral filter via a Vandermonde system, yielding a determined interpolation rather than a genuine approximation. Moreover, we point out that a $k$th-order polynomial exactly recovers $k$-hop message passing, explaining the effectiveness of spectral GNNs as an equivalence to simpler MPNNs.

[1]Anonymous Institution, Anonymous City, Anonymous Region, Anonymous Country. Correspondence to: Anonymous Author <anon.email@domain.com>.

Preliminary work. Under review by the International Conference on Machine Learning (ICML). Do not distribute.

**Section 4:** We show that the commonly cited low-/high-pass behavior of GNN spectral filters can be justified via message passing; in contrast, a Graph Fourier Transform-based argument by itself cannot establish this filtering claim.

**Section 5:** We analyze two prominent spectral GNNs for directed graphs—MagNet (Zhang et al., 2021a) and HoloNet (Koke & Cremers, 2024). Their strong empirical results stem not from spectral mechanisms, but from implementation bugs that reduce them to much simpler MPNNs.

**Scope and Terminology** We focus on node classification, primarily on general graphs (with directed and undirected graphs as the special case where each edge is accompanied by its reverse). Let $\mathcal{G} = (\mathbb{V}, \mathcal{E})$ be a graph with $n = |\mathbb{V}|$ nodes. Node features are stored in $\boldsymbol{X} \in \mathbb{R}^{n \times f}$, and node labels are $y_i \in \{1, \ldots, c\}$. The adjacency matrix is $\boldsymbol{A} \in \{0,1\}^{n \times n}$ with $\boldsymbol{A}_{ij} = 1$ indicating an edge $i \rightarrow j$. Let $\boldsymbol{D}$ denote a degree matrix (specified as in-/out-degree when needed). For undirected graphs, the Laplacian is $\boldsymbol{L} = \boldsymbol{D} - \boldsymbol{A}$, and the normalized Laplacian is $\hat{\boldsymbol{L}} = \boldsymbol{I} - \boldsymbol{D}^{-1/2} \boldsymbol{A} \boldsymbol{D}^{-1/2}$. We use $\hat{\boldsymbol{A}}$ to denote a normalized adjacency (defined in context).

## 2. Graph Fourier Basis

### 2.1. Background

Spectral graph learning builds upon (i) Graph Fourier Basis as the conceptual foundation and (ii) polynomial approximations (e.g., Chebyshev, Bernstein) as a practical surrogate to avoid eigen-decomposition (Guo et al., 2025). A Graph Fourier Transform (GFT) is typically defined through the eigen-decomposition of the normalized graph Laplacian $\hat{\boldsymbol{L}} = \boldsymbol{I} - \hat{\boldsymbol{A}}$:

$$\hat{\boldsymbol{L}} = \boldsymbol{U}\boldsymbol{\Lambda}\boldsymbol{U}^\top, \tag{1}$$

where $\boldsymbol{U}$ contains orthonormal eigenvectors and $\boldsymbol{\Lambda}$ is diagonal. $\hat{\boldsymbol{A}}$ is normalised adjacency matrix.

### 2.2. Graph Fourier Basis $\neq$ Fourier Basis of Graph

While $\boldsymbol{U}$ indeed forms an orthonormal basis and defines a valid transform for the *operator* $\boldsymbol{L}$, identifying it as the graph's own Fourier basis is conceptually misguided.

The Fourier transform was originally developed for one-dimensional signals (e.g., vibrating strings and time sequences) (Stein, 2003). Within Euclidean domains, the notion extends naturally to higher dimensions. For example, for images on $\mathbb{Z}^2$, the 2D Fourier transform represents an image as a superposition of sinusoidal patterns that vary along the horizontal and vertical directions of the grid (Russ & Neal, 2016; Gonzalez & Woods, 2017). More fundamentally, Fourier modes in Euclidean settings arise as eigenfunctions (characters) of the translation group; consequently,

"frequency" is tied to translation invariance and to sinusoidal patterns that are globally meaningful across the domain. On discrete domains the same principle holds: the discrete Fourier transform diagonalizes the action of cyclic shifts on $\mathbb{Z}_n$, and its modes are characters that represent globally meaningful oscillations on a regular grid (Oppenheim et al., 1999).

For general graphs, however, there is no canonical notion of translation and no globally consistent set of orthogonal directions; thus, the classical semantics of frequency do not directly carry over (Ricaud et al., 2019). Recent discussions further stress that the semantic properties that make Fourier analysis powerful in Euclidean settings may not be preserved when this terminology is transferred to arbitrary graphs (Guo et al., 2025).

Using Laplacian eigenvectors as the graph's "Fourier basis" is analogous to the following construction in the image domain: rather than applying the standard 2D Fourier transform—whose modes are global sinusoids aligned with the horizontal and vertical directions—one instead diagonalizes a pixel-connectivity Laplacian and treats its eigenvectors as "spectral components." This procedure indeed produces an orthonormal basis and defines a valid transform for that Laplacian. However, it is conceptually distinct from the classical Fourier basis, which is induced by the grid's translation structure. Consequently, the resulting coefficients do not, in general, admit the classical interpretation of a "spectrum" tied to translation-invariant oscillations. This is a pitfall: Graph Fourier Basis $\neq$ Fourier basis of graph.

Recent work (Guo et al., 2025) reaches a similar conclusion but misattributes the issue to continuous-vs-discrete mismatch, overlooking the well-established Discrete Fourier Transform (Oppenheim et al., 1999). The true gap lies in graphs' lack of translation structure.

## 3. Polynomial Approximations

### 3.1. Background

While interpreting Laplacian eigenvectors as the graph's "Fourier basis" remains conceptually debatable, the subsequent use of polynomial approximations raises additional issues even granting the Graph Fourier Basis interpretation.

Spectral GNNs take node features $\boldsymbol{X} \in Rnxd$ as input signals on d channels, map them to the spectral domain via the Graph Fourier Transform, apply a spectral modulation $g(\lambda)$, and then return to the spatial domain using the inverse transform (Guo & Wei, 2023), as shown in Figure 1.

To avoid the eigendecomposition of Equation 1, the filter $g(\lambda)$ is typically approximated by a polynomial $h(\lambda) = \sum_{k=0}^{K} \alpha_k \lambda^k$, so that the filtered signal can be implemented using powers of the normalized Laplacian (Kipf & Welling,

*Figure 1.* Graph Fourier transform and its inverse, showing spectral modulation by a diagonal filter in the Fourier domain.

2016; Wang & Zhang) as follows:

$$\boldsymbol{X}^* \approx \sum_{k=0}^{K} a_k \hat{\boldsymbol{L}}^k \boldsymbol{X}. \tag{2}$$

### 3.2. Pitfalls of Truncated Polynomial Approximations

Consider an "ideal" spectral modulation $\boldsymbol{\theta}^* = (\theta_1^*, ..., \theta_n^*)$ on a graph with eigenvalues $\lambda_1, ..., \lambda_n$. Writing the polynomial constraints

$$\theta_i^* = \sum_{k=0}^{K} \alpha_k \lambda_i^k, i = 1, ..., n, \tag{3}$$

yields a linear system in the coefficients $\alpha_0, ..., \alpha_K$. When the eigenvalues are distinct, choosing $K = n - 1$ gives a square Vandermonde system that determines a unique coefficient vector $\boldsymbol{\alpha} = (\alpha_0, ..., \alpha_K)$ interpolating $\boldsymbol{\theta}^*$ exactly; taking $K \geq n$ only introduces unnecessary underdetermination without improving the fit on this fixed graph. In this sense, an "infinite-hop" polynomial is not required to realize an arbitrary spectral filter on a finite graph.

Once $\boldsymbol{\alpha}$ is fixed, each coefficient $\alpha_k$-whether multiplying $\lambda^0$ or $\lambda^{n-1}$-contributes equally to matching $\boldsymbol{\theta}^*$ on the spectrum.

This reveals that the recent surge of truncated polynomial bases inadvertently favors low-degree coefficients $\alpha_0, \alpha_1, ...$ (corresponding to local neighborhood aggregation) underutilizing higher-degree coefficients.

Simply proposing new polynomial families—such as Chebyshev expansions (Defferrard et al., 2016), first-order Chebyshev expansion (Kipf & Welling, 2016), Bernstein bases

(He et al., 2021), or others (Xu et al., 2024)—amounts to blindly exploring different values of $\alpha_k$. For example, Cheb-Net (Defferrard et al., 2016), effectively sets $\alpha_0$=1, $\alpha_1$=1, while GCN (Kipf & Welling, 2016) corresponds to $\alpha_0$=2, $\alpha_1$=-1 (after renormalization). These remain particular solutions within the full interpolation space, not structurally privileged choices.

Furthermore, Equation 3 is dataset-specific: $\boldsymbol{\Lambda}$ and $\boldsymbol{\theta}$ vary across graphs. Spectral GNNs thus imply dataset-specific models, undermining their claimed generality.

### 3.3. What Polynomial Approximation Truly Leads To

Despite conceptual issues with the Fourier basis interpretation (Guo et al., 2025) and polynomial approximation pitfalls, spectral GNNs achieve strong empirical performance by reducing to multi-hop message passing. Different polynomial filters correspond to different weighted combinations of spatial aggregations, while the polynomial order directly controls the maximum hop distance of information propagation (Balcilar et al., 2021).

Polynomial filters in Equation 2—where $\hat{\boldsymbol{L}}^k \boldsymbol{X}$ aggregates $k$-hop neighborhood information (Jiang et al., 2025b)—explicitly correspond to spatially localized, weighted combinations of neighborhood aggregation that avoid eigendecomposition altogether (Balcilar et al., 2021). Classic examples include ChebNet's Chebyshev polynomials (Defferrard et al., 2016) and GCN's first-order approximation (Kipf & Welling, 2016), both of which yield bounded-hop spatial computations despite their spectral motivation.

The limitations of spectral GNNs are not limited to the pitfalls discussed in Sections 2 and 3. In Appendix A.1, we discuss additional reasons why spectral GNNs are unsuitable for node classification, and we provide simple, intuitive insights illustrating why they are ill-suited to this task.

## 4. Low-Pass and High-Pass Filters

In the discrete-time Fourier setting, consider the linear time-invariant (LTI) system

$$y[n] = x[n] + x[n - 1].$$

For a complex exponential input $x[n] = e^{i\omega n}$, the output has the form $y[n] = H(\omega)e^{i\omega n}$, where the frequency response is

$$H(\omega) = 1 + e^{-i\omega}.$$

Its magnitude is largest at low $\omega$ and smallest at high $\omega$, hence the system is *low-pass* (Oppenheim et al., 1999). A detailed derivation is provided in Appendix A.2. Similarly, the system

$$y[n] = x[n] - x[n - 1]$$

has frequency response $H(\omega) = 1 - e^{-i\omega}$, whose magnitude is smallest at low $\omega$ and largest at high $\omega$, and is therefore *high-pass*. Intuitively, $x[n] + x[n-1]$ emphasizes slowly varying (smooth) components, while $x[n] - x[n-1]$ emphasizes rapid changes.

## 4.1. GCN as a Low-Pass Filter

Analogous notions of low-pass and high-pass filtering arise as spectral filters (Balcilar et al., 2021; Liu et al., 2025). It is widely recognized that graph convolutional networks (GCNs) exhibit a low-pass bias (Wu et al., 2019; Chen et al., 2024), often attributed to aggregation with self-loops (i.e., using $\tilde{A} = A + I$). However, a commonly cited justification based solely on graph Laplacian eigenvalue ordering does not by itself establish a *filtering* claim.

Wu et al. (2019) argue as follows. After adding self-loops, the eigenvalues of

$$\tilde{L} = I - \tilde{D}^{-1/2}\tilde{A}\tilde{D}^{-1/2} \quad \text{and} \quad L = I - D^{-1/2}AD^{-1/2}$$

(with $\tilde{D}$ and $D$ the degree matrices of $\tilde{A} = A + I$ and $A$, respectively) satisfy an ordering relationship; in particular, they prove that the largest eigenvalue of the normalized Laplacian decreases after adding self-loops. Based on this spectral shrinkage, they interpret GCNs with self-loops as exhibiting a low-pass-type behavior.

However, this reasoning does not by itself establish a *filtering* claim. The above result is a statement about the *spectrum of the Laplacian*. By contrast, to call an operator "low-pass" one must characterize its *frequency response*, i.e., how the *propagation operator* scales Fourier modes across Laplacian eigenvalues (preferentially amplifying low-frequency modes and attenuating high-frequency modes). An ordering or shrinkage of Laplacian eigenvalues alone does not imply that the corresponding GCN propagation operator has maximal gain on low-frequency components.

We agree with the qualitative conclusion that GCNs are low-pass biased, but we contend that it does not follow directly from Laplacian eigenvalue ordering. Intuitively, using $A + I$ promotes feature similarity among neighboring nodes and thus favors smooth node signals. In this sense, the pair $(I + A)$ and $(I - A)$ is analogous to the classical low-pass and high-pass constructions in the discrete-time Fourier setting.

## 4.2. A Hop-Domain View: Shift-Invariant Filtering on Graphs

We now introduce a hop-indexed viewpoint that parallels the classical theory of linear time-invariant (LTI) systems in discrete-time signal processing. The key difference is that the index $n$ below is *not physical time*, but the *hop depth* of message passing. Concretely, let $S$ denote a fixed graph

propagation (shift) operator (e.g., $S = \tilde{D}^{-1/2}\tilde{A}\tilde{D}^{-1/2}$ for GCN, or another choice depending on the model). Given node features $X$, we define a hop-indexed sequence (Jiang et al., 2025b)

$$X[0] = X, \qquad X[n] = S^n X \ \ (n \geq 1),$$

so that $X[n]$ aggregates information from $n$ successive propagations.

Under this interpretation, we can study *linear shift-invariant* operators acting on the discrete index $n$ (the hop index), in direct analogy with discrete-time LTI systems. In particular, consider the hop-domain operators

$$Y[n] = X[n] + X[n-1], \tag{4}$$

and

$$Y[n] = X[n] - X[n-1]. \tag{5}$$

These operators are linear and *shift-invariant with respect to $n$*: shifting the input sequence $\{X[n]\}$ along the hop index results in the same shift of the output sequence $\{Y[n]\}$. Hence, they admit the standard frequency-response interpretation from discrete-time LTI theory, with the hop index playing the role of the discrete-time variable.

Moreover, this hop-domain construction yields an immediate interpretation of common GNN updates. For example, the GCN propagation $(I + S)X$ can be written as

$$(I + S)X = X[0] + X[1],$$

which is exactly the $n = 1$ instance of the hop-domain smoothing operator in Equation 4. Thus, a single message-passing step with a residual/self-loop term implements a first-order hop-domain smoothing operation, providing an alternative and transparent route to the widely observed low-pass bias of GCN-type layers.

Importantly, this argument is conceptually different from reasoning based on the Graph Laplacian Transform; the hop-domain view characterizes low-/high-pass behavior directly through shift-invariant operations on propagation depths.

In sum, the highly cited claim that "spectral filters" are low-pass or high-pass cannot be deduced from Graph Laplacian Transform-based spectral GNN theories; rather, it follows from message passing, viewed as hop-invariant filtering analogous to time-invariant filtering.

# 5. Case Study: Popular spectral GNNs for Directed Graphs

**MPNN is the reason for GCN's success**   Among spectral GNN models for undirected graphs, GCN (Kipf & Welling, 2016) has become one of the classical and powerful baselines (Luo et al., 2024). Although it claims to be based on

a first-order Chebyshev polynomial approximation of spectral convolutions, the actual formulation jumps directly to setting $\alpha_0 = 2$ and $\alpha_1 = -1$ (after renormalization). Since a first-order Chebyshev polynomial would give $\alpha_0 = 1$ and $\alpha_1 = 1$, strictly speaking, GCN is not first-order Chebyshev polynomial approximation of spectral convolutions, and thus cannot be viewed as a purely spectral GNN. This coefficient choice is not supported by the spectral-GNN theory itself; instead, its effectiveness is best explained through a spatial-domain perspective: 1-hop neighborhood aggregation with self-loops (Balcilar et al., 2021).

**Directed spectral GNNs: MagNet and HoloNet** We now examine two of the most popular spectral GNNs for directed graphs—MagNet (Zhang et al., 2021a) and HoloNet (Koke & Cremers, 2024). Despite their theoretical designs rooted in spectral operators for directed Laplacians, we reveal that their strong empirical performance stems from implementation flaws. These coding deviations contradict the intended spectral formulations and, in practice, transform the models into conventional spatial aggregators. This mirrors the phenomenon observed in GCN, where the "spectral" label masks a fundamentally message-passing architecture.

### 5.1. MagNet = GraphSAGE with GCN normalisation

#### 5.1.1. SPECTRAL GNN FOR DIRECTED GRAPHS

The adjacency matrix of a directed graph (Tong et al., 2020) is asymmetric, which prevents direct spectral eigendecomposition of its Laplacian. While this should reveal the incompatability of Spectral Graph Theory (which deals with identical nodes and bidirected edges) with modern graph learning, MagNet (Zhang et al., 2021a) push it forward by introducing a complex-valued Hermitian Laplacian: directionality is encoded via complex phases while the matrix remains Hermitian.

The adjacency matrix of a directed graph is generally asymmetric (Tong et al., 2020), so the associated Laplacian is non-symmetric and does not admit the standard real orthogonal spectral eigendecomposition that underpins classical spectral graph theory. This is not a minor technicality: it reflects a structural mismatch between spectral graph theory—developed for undirected graphs with bidirected edges and identical nodes—and modern graph learning, where edges are often directed and node features are diverse. Rather than treating this mismatch as a signal to move beyond the spectral paradigm, MagNet (Zhang et al., 2021a) attempts to preserve it by introducing a complex-valued Hermitian Laplacian, encoding edge direction through complex phases while maintaining Hermitian symmetry to recover a spectral decomposition.

As a result, its eigenvalues are real and eigenvectors form an orthonormal basis, enabling a spectral-style framework for directed graphs. In this section, we show that MagNet is mathematically equivalent to GraphSAGE (Hamilton et al., 2017) equipped with GCN-style normalisation.

#### 5.1.2. HERMITIAN CONSTRUCTION IN MAGNET

For a directed graph, the adjacency matrix $\boldsymbol{A}$ is generally not equal to its transpose $\boldsymbol{A}^\top$. MagNet (Zhang et al., 2021a) first symmetrises the adjacency via

$$\boldsymbol{A}_s = \frac{1}{2}(\boldsymbol{A} + \boldsymbol{A}^\top),$$

then applies GCN normalisation to obtain $\widetilde{\boldsymbol{A}}_s$. Directionality is reintroduced through an element-wise complex phase matrix

$$\boldsymbol{\Theta} = e^{2\pi q j(\boldsymbol{A} - \boldsymbol{A}^\top)},$$

where $q \in \mathbb{R}$ controls the phase magnitude and $j^2 = -1$. The resulting complex adjacency used by MagNet is

$$\widehat{\boldsymbol{A}}_s = \widetilde{\boldsymbol{A}}_s \odot \boldsymbol{\Theta} = \widetilde{\boldsymbol{A}}_s \odot e^{2\pi q j(\boldsymbol{A} - \boldsymbol{A}^\top)}$$
$$= \widetilde{\boldsymbol{A}}_s \odot \cos\big(2\pi q(\boldsymbol{A} - \boldsymbol{A}^\top)\big) + j\,\widetilde{\boldsymbol{A}}_s \odot \sin\big(2\pi q(\boldsymbol{A} - \boldsymbol{A}^\top)\big),$$

where $\odot$ denotes element-wise multiplication. Equivalently, $\widehat{\boldsymbol{A}}_s$ admits the decomposition

$$\widehat{\boldsymbol{A}}_s = \Re(\widehat{\boldsymbol{A}}_s) + j\,\Im(\widehat{\boldsymbol{A}}_s),$$

with

$$\Re(\widehat{\boldsymbol{A}}_s) = \widetilde{\boldsymbol{A}}_s \odot \cos\big(2\pi q(\boldsymbol{A} - \boldsymbol{A}^\top)\big),$$
$$\Im(\widehat{\boldsymbol{A}}_s) = \widetilde{\boldsymbol{A}}_s \odot \sin\big(2\pi q(\boldsymbol{A} - \boldsymbol{A}^\top)\big).$$

Each entry of $\Re(\widehat{\boldsymbol{A}}_s)$ and $\Im(\widehat{\boldsymbol{A}}_s)$ takes a value in $\{1, 0, \cos\alpha, \sin\alpha\}$, where $\alpha = 2\pi q$, as shown in Table 1.

As a result, $\Re(\widehat{\boldsymbol{A}}_s)$ — denoted $\overline{\boldsymbol{A}}_s$ — is a real symmetric matrix matching $\widetilde{\boldsymbol{A}}_s$ on bidirectional edges, while for unidirectional edges the corresponding entries are scaled by a factor of $\cos\alpha$ relative to those of $\widetilde{\boldsymbol{A}}_s$. $\Im(\widehat{\boldsymbol{A}}_s)$ is skew-symmetric, and can be written as

$$\Im(\widehat{\boldsymbol{A}}_s) = \tfrac{1}{2}\sin\alpha\, \boldsymbol{D}^{-\frac{1}{2}}(\boldsymbol{A} - \boldsymbol{A}^\top)\boldsymbol{D}^{-\frac{1}{2}}, \tag{6}$$

where $\boldsymbol{D}$ is the degree matrix of the graph.

#### 5.1.3. HERMITIAN PROPAGATION AND OUTPUT DERIVATION

In MagNet, node features are initialised as $\widehat{\boldsymbol{X}} = \boldsymbol{X} + j\boldsymbol{X}$. However, when following the ChebNet recursion (Defferrard et al., 2016), a coding error was introduced: an extra subtraction of $\boldsymbol{I}$ occurred when constructing the Laplacian $\hat{\boldsymbol{L}} = \boldsymbol{I} - \widehat{\boldsymbol{A}}_s$. This caused a mistaken recurrence:

$$\widetilde{\boldsymbol{T}_{k+2}} = 2\widetilde{\boldsymbol{L}}\boldsymbol{T}_{k+1} - \boldsymbol{T}_k, \text{ where } \widetilde{\boldsymbol{L}} = \boldsymbol{I} - \widehat{\boldsymbol{A}}_s - \boldsymbol{I} = -\widehat{\boldsymbol{A}}_s \tag{7}$$

*Table 1.* Case enumeration of the elements with entry $mn$ in adjacency matrices for different edge types between node $m$ and node $n$. Here, $\boldsymbol{A}_s$ is the symmetrized adjacency matrix, $\widetilde{\boldsymbol{A}}_s$ is its GCN-normalized adjacency matrix, and $\widehat{\boldsymbol{A}}_s$ is the complex-valued adjacency matrix used in MagNet with parameter $\alpha = 2\pi q$. The variable $d$ denotes the node degree.

| **MagNet** ($\alpha = 2\pi q$) | | | | | |
|---|---|---|---|---|---|
| **Edges** | $\boldsymbol{A}_s$ | $\widetilde{\boldsymbol{A}}_s$ | $\widehat{\boldsymbol{A}}_s$ | $\Re(\widehat{\boldsymbol{A}}_s)$ | $\Im(\widehat{\boldsymbol{A}}_s)$ |
| $\boldsymbol{m} \rightarrow \boldsymbol{n}$ | 0.5 | $\frac{0.5}{d}$ | $\frac{0.5}{d}e^{j\alpha}$ | $\frac{0.5}{d}\cos\alpha$ | $\frac{0.5}{d}\sin\alpha$ |
| $\boldsymbol{n} \rightarrow \boldsymbol{m}$ | 0.5 | $\frac{0.5}{d}$ | $\frac{0.5}{d}e^{-j\alpha}$ | $\frac{0.5}{d}\cos\alpha$ | $-\frac{0.5}{d}\sin\alpha$ |
| $\boldsymbol{m} \leftrightarrow \boldsymbol{n}$ | 1 | $d^{-1}$ | $d^{-1}$ | $d^{-1}$ | 0 |
| $\boldsymbol{m} \not\leftrightarrow \boldsymbol{n}$ | 0 | 0 | 0 | 0 | 0 |

rather than the correct Chebyshev form:

$$\widetilde{\boldsymbol{T}_{k+2}} = 2\hat{\boldsymbol{L}}\boldsymbol{T}_{k+1} - \boldsymbol{T}_k \tag{8}$$

When $K = 1$, the Hermitian output is as follows:

$$\begin{aligned}
\boldsymbol{Z}_1 &= \sigma(\hat{\boldsymbol{T}}_1 \boldsymbol{X}\boldsymbol{W}_{11}) = \sigma\big((\boldsymbol{I} + j\boldsymbol{0})(\boldsymbol{X} + j\boldsymbol{X})\boldsymbol{W}_{11}\big) \\
&= \sigma\big((\boldsymbol{X} + j\boldsymbol{X})\boldsymbol{W}_{11}\big) = \sigma(\boldsymbol{X}) + j\sigma(\boldsymbol{X}\boldsymbol{W}_{11}),
\end{aligned} \tag{9}$$

where $\boldsymbol{W}_{11}$ is a learnable weight matrix, and $\sigma(\cdot)$ is a non-linear activation function; in MagNet, $\sigma(\cdot) = \text{ReLU}(\cdot)$. For simplicity, we omit $\sigma(\cdot)$ in the following derivations, as this does not affect the final conclusion of our proof (Hornik, 1991; Hornik et al., 1989; Jiang et al., 2025a; Xu et al., 2019; Wu et al., 2019).

When $K = 2$, substituting $\boldsymbol{Z}_1$ from Equation 9 into the $K = 2$ update, the complex-valued output becomes:

$$\begin{aligned}
\boldsymbol{Z}_2 &= \sigma\big(\boldsymbol{Z}_1 + (\Re(\widetilde{\boldsymbol{T}_2}) + j\Im(\widetilde{\boldsymbol{T}_2}))(\boldsymbol{X} + j\boldsymbol{X})\boldsymbol{W}_{21}\big) \\
&= \sigma\big(\boldsymbol{Z}_1 + (-\Re(\widehat{\boldsymbol{A}}_s) - j\Im(\widehat{\boldsymbol{A}}_s))(\boldsymbol{X} + j\boldsymbol{X})\boldsymbol{W}_{21}\big) \\
&= \sigma\big(\boldsymbol{X}\boldsymbol{W}_{11} + \Im(\widehat{\boldsymbol{A}}_s)\boldsymbol{X}\boldsymbol{W}_{21} - \Re(\widehat{\boldsymbol{A}}_s)\boldsymbol{X}\boldsymbol{W}_{21} \\
&\quad + j\big(\boldsymbol{X}\boldsymbol{W}_{11} - \Re(\widehat{\boldsymbol{A}}_s)\boldsymbol{X}\boldsymbol{W}_{21} - \Im(\widehat{\boldsymbol{A}}_s)\boldsymbol{X}\boldsymbol{W}_{21}\big),
\end{aligned} \tag{10}$$

where $\boldsymbol{W}_{21}$ is also a learnable weight matrix.

In the end, MagNet concatenates the real and imaginary parts of the Hermitian output.

For $K = 1$, by Equation 9 we have $\Re(\boldsymbol{Z}_1) = \Im(\boldsymbol{Z}_1) = \boldsymbol{X}\boldsymbol{W}_{11}$. Substituting into the readout yields

$$\begin{aligned}
Out_1 &= \sigma\big(\Re(\boldsymbol{Z}_1)\boldsymbol{W}_{01} + \Im(\boldsymbol{Z}_1)\boldsymbol{W}_{02}\big) \\
&= \sigma\big(\boldsymbol{X}\boldsymbol{W}_{11}\boldsymbol{W}_{01} + \boldsymbol{X}\boldsymbol{W}_{11}\boldsymbol{W}_{02}\big) \\
&= \sigma\big(\boldsymbol{X}\boldsymbol{W}_1 + \boldsymbol{X}\boldsymbol{W}_2\big),
\end{aligned} \tag{11}$$

where $\boldsymbol{W}_1 = \boldsymbol{W}_{11}\boldsymbol{W}_{01}$ and $\boldsymbol{W}_2 = \boldsymbol{W}_{11}\boldsymbol{W}_{02}$ are learnable weight matrices.

For $K = 2$, based on Equation 10, we obtain:

$$\begin{aligned}
Out_2 &= \sigma\big(\Re(\boldsymbol{Z}_2)\boldsymbol{W}_{22} + \Im(\boldsymbol{Z}_2)\boldsymbol{W}_{23}\big) \\
&= \sigma\Big(\big(\boldsymbol{X}\boldsymbol{W}_{11} + \Im(\widehat{\boldsymbol{A}}_s)\boldsymbol{X}\boldsymbol{W}_{21} - \Re(\widehat{\boldsymbol{A}}_s)\boldsymbol{X}\boldsymbol{W}_{21}\big)\boldsymbol{W}_{22} + \\
&\quad \big(\boldsymbol{X}\boldsymbol{W}_{11} - \Re(\widehat{\boldsymbol{A}}_s)\boldsymbol{X}\boldsymbol{W}_{21} - \Im(\widehat{\boldsymbol{A}}_s)\boldsymbol{X}\boldsymbol{W}_{21}\big)\boldsymbol{W}_{23}\Big) \\
&= \sigma\big(\boldsymbol{X}\widehat{\boldsymbol{W}}_1 + \Re(\widehat{\boldsymbol{A}}_s)\boldsymbol{X}\widehat{\boldsymbol{W}}_2 + \Im(\widehat{\boldsymbol{A}}_s)\boldsymbol{X}\widehat{\boldsymbol{W}}_3\big),
\end{aligned} \tag{12}$$

where

$$\begin{aligned}
\widehat{\boldsymbol{W}}_1 &:= \boldsymbol{W}_{11}(\boldsymbol{W}_{22} + \boldsymbol{W}_{23}), \\
\widehat{\boldsymbol{W}}_2 &:= -\boldsymbol{W}_{21}(\boldsymbol{W}_{22} + \boldsymbol{W}_{23}), \\
\widehat{\boldsymbol{W}}_3 &:= \boldsymbol{W}_{21}(\boldsymbol{W}_{22} - \boldsymbol{W}_{23})
\end{aligned}$$

are all learnable weight matrices.

Substituting Equation 6, Equation 12 can further be simplified to :

$$\begin{aligned}
Out_2 = \sigma\big(&\boldsymbol{X}\widehat{\boldsymbol{W}}_1 + \boldsymbol{D}^{-\frac{1}{2}}\overline{\boldsymbol{A}_s}\boldsymbol{D}^{-\frac{1}{2}}\boldsymbol{X}\widehat{\boldsymbol{W}}_2 \\
&+ \tfrac{1}{2}\sin\alpha\,\boldsymbol{D}^{-\frac{1}{2}}(\boldsymbol{A} - \boldsymbol{A}^\top)\boldsymbol{D}^{-\frac{1}{2}}\boldsymbol{X}\widehat{\boldsymbol{W}}_3\big).
\end{aligned} \tag{13}$$

In MagNet's Chebyshev-polynomial implementation, the polynomial order is set to $K = 2$ (Zhang et al., 2021b); therefore, Equation 13 gives the layer-wise output of Mag-Net.

### 5.1.4. RELATIONSHIP TO GRAPHSAGE

*Table 2.* Classification accuracy (%) of Dir-GNN (Rossi et al., 2024) and MLP with different feature configurations and normalization schemes on Telegram datasets. Feature configurations include: original node features from datasets (Origin Feature), constant features (No Feature, all set to 1), and node degree variants (in-degree, out-degree, or both). **Bold** value indicates learning failure with row normalization and no features. Underline value indicates that predictions can be made accurately using only node degrees by MLP without message passing.

| Telegram | MLP | None | Row | Sym | Dir |
|---|---|---|---|---|---|
| Origin | 38.0±7.2 | 95.6±2.8 | 74.2±5.5 | 93.0±4.1 | 92.8±4.7 |
| No Feature | 38.0±0.0 | 95.4±4.0 | **38.0±0.0** | 93.0±4.7 | 93.0±3.0 |
| In-degree | 64.0±5.7 | 95.8±3.8 | 80.8±4.1 | 92.6±3.3 | 94.4±2.1 |
| Out-degree | 63.0±5.7 | 96.4±2.5 | 78.4±5.9 | 90.6±7.4 | 93.6±4.5 |
| Both degrees | 89.4±5.8 | 95.0±3.4 | 80.0±4.3 | 93.4±2.1 | 94.4±4.3 |

GraphSAGE's layer-wise update is

$$\sigma(\boldsymbol{X}\widetilde{\boldsymbol{W}}_1 + \boldsymbol{D}^{-1}\boldsymbol{A}\boldsymbol{X}\widetilde{\boldsymbol{W}}_2), \tag{14}$$

where $\widetilde{W_1}$ and $\widetilde{W_2}$ are learnable weight matrices.

Comparing with Equation 13, the first two terms correspond exactly to GraphSAGE, except that GCN's symmetric normalisation replaces GraphSAGE's row normalisation. The third term computes the difference between aggregated features from in-neighbors and out-neighbors, i.e., direction imbalance. This term contributes little in node classification as MagNet performs poorly on heterophilic graphs (Rossi et al., 2024), thus the best performance occurs when $\sin \alpha \approx 0$.

Experimentally, MagNet's performance closely mirrors that of GraphSAGE across most datasets, as reported in its original paper (Zhang et al., 2021a). The only notable exception is the Telegram dataset introduced in the MagNet study.

To better understand this case, we conducted additional experiments on the Telegram dataset with Dir-GNN (Rossi et al., 2024) and MLP. As shown in Table 2, symmetric ("Sym") normalization yields over 90% accuracy across all feature sets, while row ("Row") normalization reduces accuracy to 38%–80%. "None" and directional ("Dir") normalizations perform strongly, exceeding 95% and 92%, respectively, underscoring Telegram's high sensitivity to normalization: row normalization impairs performance.

Notably, a simple MLP achieves 89.4% accuracy using only node degree as a feature, while Dir-GNN (Rossi et al., 2024) with row normalisation attains only 38%. This suggests that the row normalisation used in GraphSAGE may suppress degree information (Jiang et al., 2025b), which helps explain MagNet's superior performance on Telegram due to its use of GCN-style normalisation.

In sum, without the aforementioned coding error, the performance of MagNet would be similar to that of ChebNet, which is worse than GraphSAGE in most cases.

### 5.2. HoloNet: Mixed Weight Coefficients

Despite its sophisticated theoretical motivation, the practical performance of MagNet (Zhang et al., 2021a) on directed graphs is underwhelming, casting doubt on whether spectral convolutions can be effectively extended to directed graphs. Building on this line of work, the more recent HoloNet model (Koke & Cremers, 2024) was proposed as a follow-up that aims to address these limitations. In particular, its ComplexFaberConv operator builds on the Hermitian Laplacian formulation introduced in MagNet (Zhang et al., 2021a) and reports improved results on several directed graph benchmarks. Although the HoloNet paper (Koke & Cremers, 2024) reports superior performance across various datasets, only the results on Chameleon and Squirrel rely on the ComplexFaberConv module (Koke, 2025). The remaining datasets are evaluated using FaberNet, which is

equivalent to multi-scale learning (Jiang et al., 2025a).

The ComplexFaberConv module in HoloNet (Koke & Cremers, 2024) is based on a flawed implementation (see the issues reported at https://github.com/ChristianKoke/HoloNets/issues). In this section, we analyze this implementation and explain the underlying reason for the unexpectedly strong performance of the flawed code.

Due to the implementation bug, ComplexFaberConv does not perform the intended multi-order aggregation proposed in HoloNet (Koke & Cremers, 2024), such as $AAXW$ or $A^\top A^\top XW$ as the order $k$ increases. Instead, only first-order propagations of the form $AXW$ or $A^\top XW$ are computed. As a result, ComplexFaberConv effectively reduces to a weighted combination of first-order aggregations with different coefficients:

$$A^\top XW_0 + 2^{-1}A^\top XW_1 + \cdots + 2^{-k}A^\top XW_k.$$

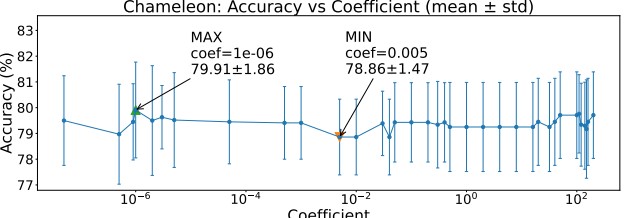

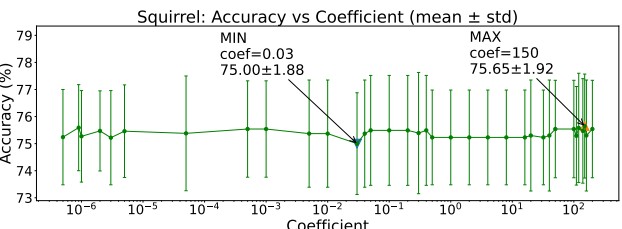

*Figure 2.* Accuracy change with different coefficients of Dir-GNN (Rossi et al., 2024) on Chameleon and Squirrel datasets. Hyperparameter choices are corresponding to the best parameters in Dir-GNN (Rossi et al., 2024).

Using $L_2$ normalisation can dramatically reduce the influence of coefficients. As shown in Appendix A.3, without $L_2$ normalisation the accuracy fluctuates drastically—from 19% to over 75% on both datasets. With $L_2$ normalisation, this sensitivity is substantially reduced: Figure 2 shows that accuracy varies only between 78.86% and 79.91%. Nevertheless, coefficients can still affect accuracy even under $L_2$ normalisation (Wang et al., 2017; 2018; Deng et al., 2022). Thus, although the fluctuations are much smaller, coefficients continue to have a non-negligible impact on model performance.

While most models adopt a fixed default coefficient (typically 1), the implementation error in ComplexFaberConv unintentionally mixes multiple coefficients, which can lead to improved empirical performance.

## 6. Conclusion

Recent criticism of spectral GNNs' Fourier basis claims (Guo et al., 2025) marks an important step forward, yet the community remains cautiously optimistic due to their empirical effectiveness. Our paper corrects their proof on graph Fourier bases, exposes a second fundamental glitch in polynomial approximations, and reveals the true source of spectral GNN success—equivalence to spatial-domain MPNNs via theoretical reduction or implementation bugs.

For undirected graphs, we show that the widely regarded "powerful" GCN is not a first-order Chebyshev spectral GNN, and its "low-pass" filtering behavior can be fully interpreted through message-passing mechanisms rather than Graph Fourier Transform-based spectral operations. For directed graphs, our analysis of MagNet and HoloNet reveals that empirical evidence no longer supports the claim that spectral methods offer distinct practical advantages.

In summary, spectral GNNs provide no unique theoretical or practical benefits. We advocate that future research in graph learning should be grounded in spatial-domain frameworks that are conceptually transparent and theoretically robust, rather than in spectral assumptions that lack rigorous justification.

## 7. Alternative Views

The community's optimism toward spectral GNNs can be seen from three aspects.

First, as discussed in Sections 2.1 and 3.1, spectral GNNs are typically built from two key ingredients: (i) graph Fourier basis and (ii) polynomial approximations. For directed graphs, a common strategy is to construct a Hermitian operator (Section 5.1.1) so that a Fourier-type basis remains available.

Second, a recent position paper (Guo et al., 2025) argues that the Fourier bases adopted by many spectral GNNs lack a clear spectral semantics. Nevertheless, the authors remain optimistic about spectral GNNs because these models often achieve strong empirical performance.

Third, spectral GNNs have been extended to directed graphs, and several widely used directed spectral architectures—such as MagNet (Zhang et al., 2021a) and HoloNet (Koke & Cremers, 2024)—have been highly influential, directly and indirectly shaping many follow-up works. More broadly, new variants of spectral GNNs continue to

appear at scale each year, reflecting sustained interest and optimism within the community.

For these reasons, we treat this prevailing optimism—either the belief that spectral GNNs genuinely "capture the graph spectrum" in a meaningful way, or the belief that spectral GNNs are effective in practice—as our alternative view. In this position paper, we focus specifically on spectral GNNs whose constructions are explicitly based on the graph Fourier basis.

## 8. Call to Action

Since spectral GNNs are theoretically flawed, there is no solid reason to expect their "spectral" mechanisms to consistently yield superior performance. When strong results appear, they are more plausibly explained by equivalences to simpler MPNNs, or by implementation/training effects rather than any meaningful spectral semantics. Therefore, we call for a more decisive shift in how the community allocates attention and evaluates new spectral claims.

- **Stop new research on spectral GNNs for node classifications.** We argue that continuing to develop increasingly complex spectral constructions is largely unnecessary and resource-consuming, especially when comparable performance can be obtained by simpler MPNNs that make the underlying mechanism explicit.

- **Be suspicious of good performance reported by spectral GNNs.** When a new spectral GNN reports strong performance, do not take the "spectral" explanation at face value. Verify reproducibility and check that the implementation is consistent with the algorithm claimed in the paper.

- **Reframe filtering in the hop domain and study hop-invariant operators.** Low- and high-pass behavior can be characterized more directly in the hop domain through depth-indexed message passing, analogous to shift-invariant filtering in linear time-invariant (LTI) systems. We encourage deeper investigation of hop-invariant formulations and their associated inductive biases, rather than relying on Graph Fourier Transform–based signal processing when its underlying assumptions are not warranted.

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

# A. Appendix

## A.1. Further Discussions

In this section, we provide further discussions highlighting fundamental problems with spectral GNNs.

**Feature-Driven Tasks**   Classical spectral graph theory (Chung, 1997; Hammond et al., 2009) assumes identical nodes, differing only in their positions within the graph, making graph connectivity the sole focus of analysis. Modern node classification tasks, however, involve graphs where node features differ substantially from node to node. Consequently, spectral notions of "low/high frequency" ignore node feature distributions, making Laplacian-based spectral intuition theoretically unsound for learning tasks on graphs with diverse node features.

**Eigenvector Instability**   Real-world graphs like citation networks tolerate minor edge/node perturbations without affecting node classification performance. Yet Laplacian eigendecomposition—particularly eigenvector rotations in near-degenerate eigenspaces—exhibits extreme sensitivity to such changes (Tao, 2008; Davis & Kahan, 1970; Hata & Nakao, 2017; Noschese & Reichel, 2019). This extreme sensitivity to minor fluctuations contradicts the stability requirements of node classification tasks and undermines spectral GNNs' reliance on eigenvectors as robust graph "frequencies." Such disconnect suggests spectral methods fundamentally mismatch practical graph learning needs.

**Polynomial Choices Are Arbitrary**   Specific polynomial families (Chebyshev in ChebNet (Defferrard et al., 2016), Bernstein in BernNet (He et al., 2021)) are presented as principled approximations, yet alternatives exist without unique theoretical justification. Notably, higher-order polynomials do not consistently improve performance (He et al., 2021). Low-order approximations like GCN (Kipf & Welling, 2016)—a first-order Chebyshev simplification—often outperform more complex spectral models like ChebNet (Defferrard et al., 2016; Zhang et al., 2021a). Even ChebNet typically uses small $K = 2$, as larger $K$ degrades performance. These observations suggest that the practical success of many "spectral" GNNs may be better explained by their induced spatial aggregation behavior (and the associated regularization/inductive bias) than by a literal Fourier interpretation.

Overall, while Laplacian eigenvectors mathematically decompose operators derived from graph connectivity, interpreting them as a classical Fourier basis with frequency semantics for the graph itself lacks justification. Spectral terminology thus provides no principled guidance for architecture design or interpretability claims about "frequencies" on graphs.

## A.2. Low Pass Filter in Discrete Domain

Consider the Linear Time-Invariant (LTI) Systems with input $x[n]$ and output

$$y[n] = x[n] + x[n-1].$$

Its impulse response is

$$h[n] = \delta[n] + \delta[n-1],$$

where $\delta[n]$ is Kronecker delta function.

The discrete-time Fourier transform (DTFT) of $h[n]$ gives the frequency response

$$H(\omega) = \sum_{n=-\infty}^{\infty} h[n]\, e^{-i\omega n} = 1 + e^{-i\omega}. \tag{15}$$

For an input $x[n] = e^{i\omega n}$, the output is

$$y[n] = H(\omega)\, e^{i\omega n},$$

so each basis function $e^{i\omega n}$ is scaled by $H(\omega)$.

**Magnitude Response**

Rewrite Equation 15, we get:

$$H(\omega) = e^{-i\omega/2} \left( e^{i\omega/2} + e^{-i\omega/2} \right) = 2\cos\left(\frac{\omega}{2}\right) e^{-i\omega/2}.$$

Thus,

$$|H(\omega)| = 2 \left| \cos\left(\frac{\omega}{2}\right) \right|.$$

**Low-Pass Property**

At $\omega = 0$, $|H(0)| = 2$    (maximum gain).

At $\omega = \pi$, $|H(\pi)| = 0$    (zero gain).

Moreover, $|H(\omega)|$ decreases monotonically from 2 to 0 as $\omega$ increases from 0 to $\pi$.

### A.3. Influence of $L_2$ Normalization on Stability

When $L_2$ normalization is applied to node features by rescaling $X$ such that each node vector has unit $L_2$ norm. normalizes each node's feature vector, which stabilizes learning and reduces sensitivity to the scale (coefficients) of the subsequent linear weights. As shown in Figure 3, without this stabilization the accuracy varies dramatically: on CHAMELEON it fluctuates from 19.58% to 79.47%, and on SQUIRREL from 19.49% to 79.22%. Wang et al. (2017) discuss the necessity of $L_2$ normalization to improve stability.

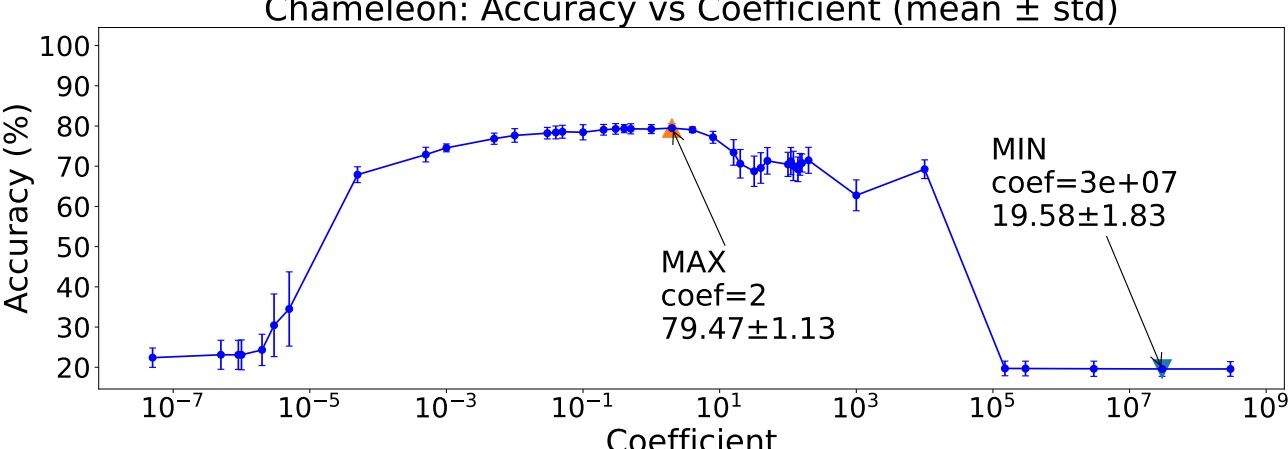

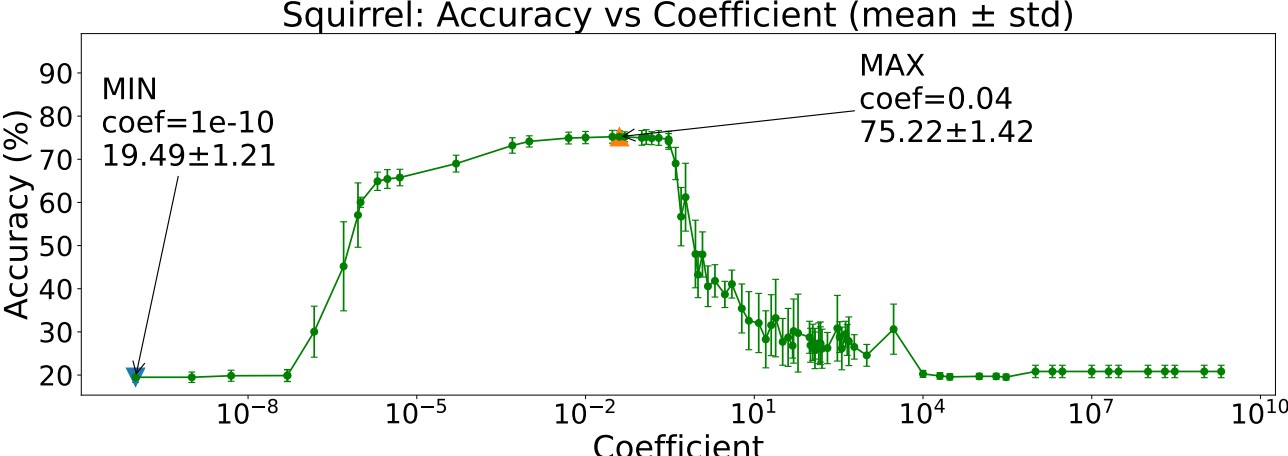

*Figure 3.* Accuracy change with different coefficients of Dir-GNN (Rossi et al., 2024) on Chameleon and Squirrel datasets. Hyperparameter choices are corresponding to the best parameters in Dir-GNN, except that without $L_2$ normalisation.

