# OpenReview forum: "Position: Spectral GNNs Are Neither Spectral Nor Superior for Node Classification"
_ICML.cc/2026/Position_Paper_Track — Submitted to ICML 2026 Position Paper Track_

### Official Review · Reviewer_Kc3R · 2026-03-09

**Significance:** 2
**Argument Clarity:** 2
**Rating:** 2
**Confidence:** 4

**Questions:**

Q1. The paper mainly analyzes GFT-based spectral GNNs, but the conclusion is stated for spectral GNNs in general. Does this also cover graph wavelet-based spectral models, or should the scope be stated more carefully?

Q2. The paper repeatedly attributes the effectiveness of spectral GNNs to their equivalence to MPNNs. Could the authors clarify more precisely in what sense this equivalence holds and under which assumptions?

Q3. The notion of “spectral semantics” in the paper seems quite narrow. How do the authors view other possible roles of spectral methods, such as multi-scale analysis, stability, or directional inductive bias?

Q4. The discussion of MagNet and HoloNet relies heavily on implementation-related issues. Without a systematic replication study, is it too strong to use this as evidence against the model principles themselves?

Q5. The paper focuses on node classification, but some conclusions read as if they apply to spectral GNNs more broadly. Could the authors clarify the intended scope of their position?

Q6. If the paper argues that current spectral GNNs are not genuinely spectral, could the authors briefly comment on what would count as a genuinely spectral graph learning method?

**Alternative Views Section:**

Yes

**Compliance With Llm Reviewing Policy A Conservative:**

Affirmed.

**Discussion Potential:**

2

**Final Justification:**

I am maintaining my original score, as the authors apparently did not address my comments in their rebuttal. Therefore, my final recommendation remains rejection.

**Paper Summary:**

The central argument of this work is that spectral graph neural networks (spectral GNNs) do not truly capture the “spectral semantics” of graphs and, importantly, do not provide clear practical advantages over spatial message-passing methods. The authors try to support this claim by closely examining the theoretical justifications commonly used to explain the success of spectral approaches and questioning whether these explanations are actually well founded. Ultimately, the paper argues that the effectiveness of many spectral GNNs arises not from a genuine frequency-domain mechanism, but largely from their equivalence to message passing neural networks (MPNNs).

**Position:**

Yes

**Position In Title:**

Yes

**Related Work:**

2

**Strengths And Weaknesses:**

Strengths:

1. The paper is generally easy to follow and clearly written.

2. The motivation to critically re-examine spectral GNNs is interesting and thought-provoking.

Weaknesses:

1. The discussion of MagNet and HoloNet mainly focuses on implementation-related issues. If the main claim is that their reported success is largely due to implementation errors, this point should ideally be supported by replication or controlled verification; otherwise, the conclusion may remain controversial.

2. The statement “Stop new research on spectral GNNs for node classification” is too strong given the current level of evidence, and would require more theoretical support and empirical validation to be fully justified.

3. The directed-graph case study seems too limited to support a broader conclusion about graph learning in general, and it is even less sufficient to conclude that spectral GNNs have no further value or necessity.

**Support:**

2

---

### Official Review · Reviewer_6ieG · 2026-03-12

**Significance:** 3
**Argument Clarity:** 4
**Rating:** 5
**Confidence:** 4

**Questions:**

In lines 153–160 the authors mathematically show that, once coefficients α are fixed by interpolation, all $α_k$ contribute equally to matching θ* on the spectrum. Could the authors provide numerical evidence (e.g., a small synthetic example or empirical plot) that illustrates the “equal contribution” of high‑degree coefficients in an optimal graph filter? Such a demonstration would help readers connect the analytical result to concrete behavior in finite graphs.

**Alternative Views Section:**

Yes

**Compliance With Llm Reviewing Policy A Conservative:**

Affirmed.

**Discussion Potential:**

3

**Paper Summary:**

This position paper argues that the eigenvectors of the graph Laplacian (the graph Fourier basis) are not equivalent to the canonical Fourier basis from Euclidean domains, due to structural differences between Euclidean domains and general graphs. The authors further claim that the practical effectiveness of GCNs is driven by message‑passing mechanisms rather than by Graph Fourier Transform–based spectral formulations. Two representative spectral GNN models are analyzed to expose fundamental pitfalls of spectral approaches.

**Position:**

Yes

**Position In Title:**

Yes

**Related Work:**

3

**Strengths And Weaknesses:**

**Strengths**

1) The theoretical analysis showing the non‑equivalence between graph Fourier bases and Euclidean Fourier bases for undirected graphs is solid and well presented.
2) Empirical analysis for directed graphs supports the theoretical claims and strengthens the paper’s overall argument.
3) The discussion and demonstration of the limitations of truncated polynomial approximations (e.g., why low‑degree truncation biases toward low‑frequency components) are insightful and practically relevant.

**Weaknesses**

I have no major concerns: the claims are convincingly demonstrated both theoretically and empirically. A few minor clarifications (see question) would further strengthen the paper.

**Support:**

4

---

> ### Author Rebuttal · Authors · 2026-03-25
>
> Dear Reviewer 6ieG,
>
> Thank you very much for endorsing our paper. We sincerely appreciate your thoughtful suggestion to strengthen our paper.
>
> As you have noted, Spectral GNNs rely on two internally flawed steps that combine to appear effective only by collapsing into the MPNN framework:
> - **Step 1**: Graph Fourier Transform
> - **Step 2**: Polynomial Approximations
>
> Our paper analyzes these steps independently:
> - **Section 2** explains that Step 1 is fundamentally flawed.
> - **Section 3** then examines Step 2 under the temporary assumption that Step 1 were valid, so that we can isolate the internal issues of polynomial approximation without conflating them with those of the Graph Fourier Transform.
>
> In Section 3, we show that Step 2 is not a classical polynomial approximation task at all, but rather a finite interpolation problem.
> This  differs from classical approximation methods like Taylor or Chebyshev expansions, where higher‑degree terms naturally refine the approximation step by step. In a Vandermonde interpolation system, this hierarchy simply doesn’t exist—every polynomial order matters equally.
> This leads to:
> >  **Deduction 1**: all $α_k$ contribute equally to matching θ* on the spectrum,
>
> Here, “equally” is meant in a comparative sense, highlighting the absence of the hierarchical refinement property present in classical approximation theory.
>
> Motivated by your question, we will clarify in the revision that **Deduction 1** contradicts empirical evidence. For example, [1] shows that aggregating messages from low-order neighborhoods consistently outperforms higher-order aggregation.
> Since Deduction 1 follows directly from the premise that the Graph Fourier Transform provides a meaningful "Fourier basis" for graph signals, its empirical failure warrants rejecting that premise.
> This further reinforces our conclusion in Section 2 that Step 1 is fundamentally flawed.
>
> In our original submission, we already used these deduction to highlight the tension between the theoretical formulation of spectral GNNs and their design practice — for instance, that the framework implies favouring low-order terms and amounts to blindly exploring polynomial coefficients. Thanks to your question, we will additionally note that these deduction also contradict empirical evidence, strengthening the argument further.
>
> By the same reasoning, all other deductions in Section 3.2 (e.g., that spectral GNNs imply dataset-specific models, undermining their claimed generality) inherit the invalidity of this premise. Taken together, these deductions expose an internal inconsistency: conclusions derived from the theoretical formulation of spectral GNNs contradict both their design practice and empirical evidence. Our intention in developing these deductions is precisely to highlight this tension within the framework.
>
> In all, our broader aim has been to expose the flaws in Spectral GNNs across multiple aspects:
> - Each of the two constituent steps is independently problematic, rooted in a misapplication of Fourier analysis and polynomial approximation respectively.
> - The design choices of spectral GNNs, such as favouring low-order terms, unprincipled parameter search, and implicit dataset-specificity that undermines their claimed generality, are inconsistent with their own theoretical formulation.
> - As your comment helpfully prompted us to clarify, Deduction 1 also contradicts empirical evidence, providing additional grounds for rejecting the underlying premise.
>
> We emphasize that the analysis of Step 1 alone is sufficient to invalidate the spectral GNN framework, as also argued in [2]; the analysis of Step 2 is offered as a complementary and independent line of critique.
>
> [1] Jiang, Q., Wang, C., Lones, M., and Pang, W. Demystifying
> MPNNs: Message passing as merely efficient matrix multiplication, 2025. URL https://arxiv.org/abs/2502.00140.
> [2] Guo, Y., Tang, H., Ma, J., Xu, H., and Wei, Z. Position:
> Spectral GNNs rely less on graph fourier basis than conceived. In Forty-second International Conference on
> Machine Learning Position Paper Track, 2025. URL https://openreview.net/forum?id=JkcSsFWdGP.
>
>
> It is truly a pleasure to have a reviewer like you, and we are grateful for the opportunity to engage with such a careful and constructive review.

---

> > ### Author Rebuttal · Reviewer_6ieG · 2026-04-04
> >
> > The authors have addressed my concern in their detailed repsonse.

---

### Official Review · Reviewer_WfQp · 2026-03-12

**Significance:** 1
**Argument Clarity:** 2
**Rating:** 1
**Confidence:** 5

**Questions:**

I suggest that the authors more carefully review the literature on signal processing on graphs and spectral graph theory to support their claims.

**Alternative Views Section:**

Yes

**Compliance With Llm Reviewing Policy A Conservative:**

Affirmed.

**Discussion Potential:**

1

**Paper Summary:**

In this paper, the authors claim that spectral GNNs are theoretically flawed with several pitfalls and suggests researchers stop new research on spectral GNNs for node classifications. Specifically, their argument is mainly supported by two claims: 1. Graph Fourier Basis $\neq$ Fourier  Basis of Graph, and 2. (n−1)th-order polynomial yields a determined interpolation rather than a genuine approximation.

**Position:**

Yes

**Position In Title:**

Yes

**Related Work:**

2

**Strengths And Weaknesses:**

Strenghs:

1. The author conduct experiments to demonstrate that the sucess of directed spectral GNNs largely arise from certain implementation tricks.

Weaknesses:

1. The main claims are not correct and convincing.

> a. Graph Fourier Basis $\neq$ Fourier Basis of Graph

The authors argue that the graph Fourier basis is not a true Fourier basis mainly because general graphs lack a global notion of translation and a shared set of orthogonal directions. However, this reasoning is not entirely convincing. In graph signal processing, the graph Fourier basis is defined individually for each graph rather than being shared across all graphs; each graph therefore has its own signal space. Moreover, translation-like operators have been defined in the graph spectral domain [1]. The authors should carefully read this work.


[1] Shuman, David I., et al. "The emerging field of signal processing on graphs: Extending high-dimensional data analysis to networks and other irregular domains." IEEE signal processing magazine 30.3 (2013): 83-98.


> b. Following spectral GNN reasoning, we show that on an n-node graph, an (n−1)th-order polynomial exactly interpolates any ideal spectral filter via a Vandermonde system, yielding a determined interpolation rather than a genuine approximation.

The authors argue that an (n−1)-th order polynomial can exactly interpolate any spectral filter via a Vandermonde system, and therefore polynomial filters correspond to interpolation rather than genuine approximation. However, this argument appears to misunderstand the role of graph filters in learning problems. In the graph Fourier domain, polynomial or Chebyshev filters simply define a basis of the function space over the graph spectrum, which in principle can represent arbitrary signals. This representational property is well known and does not contradict the use of polynomial filters. More importantly, in node classification we do not observe the full signal on the graph; only a small subset of node labels is available. The learning problem is therefore not to interpolate a predefined spectral filter, but to learn a model that generalizes from partially observed labels to unlabeled nodes.

> c. A Hop-Domain View: Shift-Invariant Filtering on Graphs

The proposed hop-domain LTI interpretation is questionable. Since $X[n] = S^n X$ is a deterministic propagation process rather than an arbitrary signal sequence, interpreting hop depth as a time index and invoking classical shift-invariant filtering is not theoretically well motivated. I also disagree with the claim that the low-pass or high-pass nature of ``spectral filters'' cannot be deduced from the graph Laplacian. Spectral GNNs and message passing are two equivalent formulations of the same propagation process: polynomial or other spectral filters correspond exactly to multi-hop message passing, whose smoothing behavior is naturally characterized by the Laplacian spectrum.

2. In the experimental section, the authors only evaluate on directed graphs. However, many spectral GNN models have demonstrated strong empirical performance on undirected graphs, such as ChebNetII.

3. The writing needs to be improved.

**Support:**

3

---

### Official Review · Reviewer_qbqE · 2026-03-13

**Significance:** 1
**Argument Clarity:** 2
**Rating:** 2
**Confidence:** 4

**Questions:**

1. Could you elaborate on the hop-domain filtering call to action. How would studying hop-invariant operators be a better alternative to spectral GNN?

**Alternative Views Section:**

Yes

**Compliance With Llm Reviewing Policy A Conservative:**

Affirmed.

**Discussion Potential:**

1

**Paper Summary:**

The paper hold a position on viewing that spectral GNN is not truly spectral in the mathematics sense nor useful for node classification. The paper showed that Laplacian eigenvectors do not generally have the properties of a true Fourier basis and the polynomial approximation view of GNN is not explanatory. The paper called to halt new research on complicated spectral filters for node classification, be doubtful of good performance reported by spectral GNNs, and advocated for studying hop-invariant operators.

**Position:**

Yes

**Position In Title:**

Yes

**Related Work:**

1

**Strengths And Weaknesses:**

# Strength

- The paper raised a valid issue in the spectral GNN research that it lacks principled formulation

# Weakness

- The paper focused too much on technicality and terms. This rigidity is best suited for theoretical works that can provide trade-offs or new design, but the targets of this position paper is the node classification problem, which is more on the practical side rather than theoretical.
- Node classification is not a topic of interest for the community at the moment.

**Support:**

1

---

### Decision · Program_Chairs · 2026-04-30

**Decision:**

Reject

**Comment:**

This work presents an interesting opinion that can be of importance to the community in terms of steering efforts away from spectral GNNs. However, several reviewers challenged this opinion. For example, some argued that the focus on the technicality of whether spectral GNNs are indeed spectral from a mathematical point of view might miss the practical benefits. In other words, even if spectral GNNs are not spectral from theoretical point of view, it may have inspired beneficial algorithms and therefore banning this direction might not be a good idea. Unfortunately, the authors decided not to respond to such comments. Hence, this position paper requires additional argumentation before being published.